# Platinum-Cobalt Nanowires for Efficient Alcohol Oxidation Electrocatalysis

**DOI:** 10.3390/ma16020840

**Published:** 2023-01-15

**Authors:** Wenwen Wang, Xinyi Bai, Xiaochu Yuan, Yumin Liu, Lin Yang, Fangfang Chang

**Affiliations:** Collaborative Innovation Centre of Henan Province for Green Manufacturing of Fine Chemicals, Key Laboratory of Green Chemical Media and Reactions, Ministry of Education, School of Chemistry and Chemical Engineering, Henan Normal University, Xinxiang 453007, China

**Keywords:** direct alcohol fuel cells, nanowire, nanocatalysts, alloy effect, lattice strain

## Abstract

The compositions and surface facets of platinum (Pt)-based electrocatalysts are of great significance for the development of direct alcohol fuel cells (DAFCs). We reported an approach for preparing ultrathin Pt_n_Co_100−n_ nanowire (NW) catalysts with high activity. The Pt_n_Co_100−*n*_ NW alloy catalysts synthesized by single-phase surfactant-free synthesis have adjustable compositions and (111) plane and strain lattices. X-ray diffraction (XRD) results indicate that the alloy composition can adjust the lattice shrinkage or expansion of Pt_n_Co_100−n_ NWs. X-ray photoelectron spectroscopy (XPS) results show that the electron structure of Pt is changed by the alloying effect caused by electron modulation in the d band, and the chemical adsorption strength of Pt is decreased, thus the catalytic activity of Pt is increased. The experimental results show that the activity of Pt_n_Co_100−n_ for the oxidation of methanol and ethanol is related to the exposed crystal surface, strain lattice and composition of catalysts. The Pt_n_Co_100−n_ NWs exhibit stronger electrocatalytic performance for both methanol oxidation reaction (MOR) and ethanol oxidation reaction (EOR). The dominant (111) plane Pt_53_Co_47_ exhibits the highest electrocatalytic activity in MOR, which is supported by the results of XPS. This discovery provides a new pathway to design high activity, stability nanocatalysts to enhance direct alcohol fuel cells.

## 1. Introduction

Direct alcohol fuel cells are gaining increased attention owing to advantages such as good durability, high energy conversion efficiency, clean and low temperature operation, environmental protection and sustainable regeneration [1,2]. Pt is the best catalyst for fuel cells and determines the properties of a fuel cell to a large extent, and is also an effective catalyst for the EOR and MOR. Huang’s team found that synthetic Pt catalyst showed very interesting dimension-dependent activity against ORR and MOR [3]. The conventional catalyst for MOR is Pt because of its excellent performance in selectivity and activity [4]; however, the use of Pt is restricted due to its high price and surface poisoning. Additionally, conserving expensive Pt catalysts is essential to advance their scalable usage in a sustainable energy economy. Pt-based catalysts are in a stage of rapid development, and the introduction of non-precious metals is notable for increasing the performance of Pt and reducing the amount of Pt required. Pt was alloyed with first-row transition metals [5,6,7,8], which is an acceptable method to solve the above-mentioned problems by virtue of the special electronic structure and geometric configuration [9,10]. One-dimensional (1D) nanostructured electrocatalysts, such as NWs, have been favored to solve the challenge with nanoparticles (NPs) [11,12,13,14]. Pt-based NWs possess significant activities [15]. CoPt nanowires have shown good performance in oxygen reduction [16,17] and methanol oxidation [18]. In addition, Pt NWs promote electron transfer during the reaction due to their larger accessible surface area [19]. Therefore, a clear pathway to enhance the performance of electrocatalysts is to develop Pt-M NWs with ultralong and ultrathin structures. Pt*_n_*Co_100−n_ NWs were prepared through a seed-mediated process in which superior nanostructures met the requirements for high density and exponential surfaces, showing better performance than NPs [20]. Using Co_25_Pt_75_ nanoparticles as catalyst, the highest reported current density is 47.1 mA cm^−2^ (Xia et al. [21]). For Co_40_Pt_60_ alloys used as catalysts in the shape of nanowires, the highest reported current density is 14 mA/cm^2^ (Bertin et al. [22]), while using Co_23_Pt_77_ NWs as catalyst, the highest MOR activity reported by Serrà et al. [23] is 7 mA cm^−2^.

In this work, we used a straightforward one-step hydrothermal technique to prepare Pt_n_Co_100−*n*_ NWs with tunable compositions, highly active facets, and lattice strain. We firstly found that the electrocatalytic performance of ultrathin Pt_n_Co_100−n_ nanowires (NWs) (≈2.1 nm) is improved by regulating the compositions, high active facets and lattice strain of catalysts for alcohol oxidation. In mass activity (MA) and specific activity (SA), Pt_n_Co_100−n_ NW catalyst demonstrates higher MOR and EOR activity and stability in comparison to Pt/C. The structure and chemical makeup of the catalysts were examined using high-resolution transmission electron microscopy (HR-TEM) and XRD. The findings demonstrated that the catalyst compositions alter the lattice strain and dominating (111) facets of Pt_n_Co_100−*n*_ NWs with adjustable compositions. The improved activity and stability for MOR and EOR of ultrathin Pt_n_Co_100−n_ NWs reveals the relation among morphology, facets, lattice strain and bimetallic compositions. Additionally, the inclusion of Co alters the crystal structure, redesigning the electrical structure and considerably reducing the consumption of Pt.

## 2. Experimental Section

### 2.1. Chemicals

Ethanol, platinum (II) acetylacetonate (Pt(acac)_2_), oleylamine, cobalt (II) acetylacetonate (Co(acac)_2_), *N*, *N*-dimethylformamide (DMF), and 1-heptanol were bought from Aladdin. Hexane and KOH were obtained from Stem Chemicals. Potassium chloride and ethylene glycol (EG) were purchased from Deen reagent. All gases were purchased from Airgas.

### 2.2. Preparation of Pt_n_Co_100−n_ NWs

The Pt_n_Co_100−n_ catalysts were prepared using a one-step hydrothermal method. In this synthesis [24], KOH (1.0 g) was entirely dispersed in a solution including 5.0 mL 1-heptanol, 10 mL DMF, 20 mL oleylamine and 36 mL EG by magnetic stirring. Then, different atomic ratios of Pt(acac)_2_ precursors and Co(acac)_2_ precursors were dissolved in the above solution, stirred overnight, and subsequently transferred to an autoclave and kept at 180 ℃ for 8 h to obtain Pt_70_Co_30_ NWs. The ratio of Co(acac)_2_ and Pt(acac)_2_ precursors was adjusted to control compositions. The Pt_n_Co_100−*n*_ NWs were washed with ethanol and collected though centrifugation and dispersed in ethanol for further use.

The as-obtained Pt_n_Co_100−*n*_ NWs were loaded on Vulcan XC-72 (20 wt% metal loading) carbon to prepare Pt_n_Co_100−*n*_/C for catalytic measurement. Vulcan XC-72 carbon was added into ethanol and stirred to form a uniform dispersion, and then the as-obtained Pt_n_Co_100−n_ NWs catalyst was added to the above dispersion and stirred to ensure the catalysts loaded on carbon, which was collected and dried to obtain the final catalyst. Pt NWs was synthesized in the same way without adding the Co(acac)_2_ precursor.

### 2.3. Characterizations

Inductively coupled plasma-optical emission spectroscopy (ICP-OES) was used to analyze the elementary compositions and loading for catalysts [25]. Transmission electron microscope (200 kV) scanning performed on an FEI Titan G2 F20 microscope was used to investigate the morphology of the resulting catalysts [26]. XRD was conducted to analyze the structures of Pt_n_Co_100−_*_n_*/C. The XPS technique was used to analyze the chemical state of the catalysts [27].

### 2.4. Electrochemical Measurements

The ink was created by dispersing 2.0 mg of catalysts using ultrasonication for 50 min in a solution including isopropanol, water, and 5 wt% Nafion (19:1:0.015, *v*/*v*/*v*). To form a uniformly thin film that served as the working electrode, the catalyst ink (10 μL) was pipetted onto a polished glassy carbon rotating disk electrode (0.196 cm^2^). On a CHI 760E electrochemical workstation, electrochemical tests were conducted in a three-electrode cell (CH Instrument, Inc., Bee Cave, TX, USA). By using cyclic voltammetry (CV) and rotating disk electrode (RDE) curves, the electrocatalytic activity of the Pt_n_Co_100−*n*_/C was investigated. Before CV and RDE tests, the electrolyte was bubbled with pure N_2_ and O_2_, respectively, for more than 30 min to build a saturated testing environment. The properties of Pt_n_Co_100−*n*_/C were assessed mainly by MA as well as SA.

## 3. Results and Discussion

### 3.1. NW Morphology

KOH, Pt(acac)_2_ and Co(acac)_2_ were added to synthesize Pt_n_Co_100−n_ nanoalloy catalysts in a mixture containing oleylamine DMF, 1-heptanol and EG, which was stirred overnight, transferred to an autoclave, and then kept at 180 ℃ to obtain the target product. 1-heptanol acted as a reducing agent of Co^2+^ to Co during the synthesis process. In terms of morphology and structure, TEM analysis showed that the as-obtained Pt_n_Co_100−n_ exhibited a bundle network structure, which was composed of multiple ultra-thin nanowires interwoven (Figure 1a–c). The diameter of each nanowire was about 0.21 nm, and the ultra-long nanowires were several micrometers in length with a high aspect ratio. The reason why Pt_n_Co_100−n_ NWs possess this special structure has two aspects. Firstly, the hydrophobic interaction of DMFs helps to minimize the free energy of Pt_n_Co_100−n_ synthesis, resulting in a strong cohesive interaction on the NWs. Secondly, there is a propensity to develop active defects along the NWs, which combine with the different NWs contacted to form the network structure [28]. Structural defects, such as grain boundaries and holes, can enhance catalytic activity. Figure 1 shows TEM images of Pt_n_Co_100−n_ NWs (Figure 1a–c) and HR-TEM observations (Figure 1d–f). The HR-TEM image shown in Figure 1 reveals the single crystal properties of each alloy nanowire. The lattice fringes of Pt_n_Co_100−n_ fell in between Pt and Co, which was gradually increased, indicating that Co atoms successfully integrated into the Pt nanostructure. As the proportion of platinum increased, the morphology of the NWs was gradually thinned and terminated by the (111) facet, which was reported in our previous work [29,30].

The structure of Pt_n_Co_100−n_/C was characterized by XRD and XPS techniques. Figure 2a displays the XRD patterns of Pt_27_Co_73_/C, Pt_53_Co_47_/C, and Pt_70_Co_30_/C. The diffraction peaks of Pt_n_Co_100−n_/C with different compositions were located between Pt (JCPDS No. 04-0802) and Co (JCPDS No. 15-0806), and were obviously exponential with face-centered cubic (fcc) PtCo alloy [31]. The peaks located at 40.37°, 39.87° and 39.82° corresponded to the (111) planes of Pt_27_Co_73_/C, Pt_53_Co_47_/C and Pt_70_Co_30_/C, respectively. The (111) diffraction peaks’ small downshift indicated a slight increase in Pt content in the NWs. The broad (111) peak, (200) peak and (220) peak indicated the formation of a nanoscale crystal structure. XRD data showed that the compositions of the nanoalloy catalysts for Pt_27_Co_73_/C, Pt_53_Co_47_/C and Pt_70_Co_30_/C were uniform, and there was no phase segregation in the samples. The diffraction peak of Pt_n_Co_100−n_/C became sharp and showed a slight blue shift with the increase in Co%, proving that the smaller Co atoms entered into the NWs and might result in lattice strain. The lattice strain would influence the electrocatalytic properties through a possible geometric effect. As shown in Figure 2b, Pt_n_Co_100−n_/C exhibited lattice expansion when the amount of Pt was less than 50%, and lattice shrinkage when Pt% was greater than 50%, conforming to Vegard’s law. XPS analysis was performed to understand the chemical states of Pt and Co as well as surface compositions of Pt and Co in Pt_n_Co_100−n_/C. Figure 2c, d shows the Pt 4f and Co 2p XPS spectra, indicating that Co and Pt were in the metallic state. XPS analysis also confirmed the existence of Pt and Co in the as-obtained Pt_n_Co_100−n_/C with different compositions. Figure 2c illustrates the Pt 4f XPS spectra for Pt_27_Co_73_/C, Pt_57_Co_43_/C and Pt_70_Co_30_/C. The peaks of Pt 4f_7/2_ appeared at 71.9, 71.9, and 72.2 eV, and the peaks of Pt 4f_5/2_ appeared at 75.1, 75.1 and 75.3 eV, respectively. The binding energy of Pt 4f showed an obviously blue shift relative to the XPS data for pure Pt, which suggested charge transfer from Pt to Co. The positive shift of the Pt binding energy suggested that the center of the d-band had shifted down.

### 3.2. Electrocatalytic Properties

The CV technique was used to study the MOR and EOR performance of the Pt_n_Co_100−n_ NWs/C as anodic catalysts. As shown in Figure 3, CVs were recorded in N_2_-saturated 0.1 M HClO_4_ + 0.5 M CH_3_OH/C_2_H_5_OH solution with a 50 mV s^−1^ scan speed. In MOR and EOR, the MA and SA were the foremost parameters. The Pt loading of the Pt_27_Co_73_/C, Pt_53_Co_47_/C, Pt_70_Co_30_/C, Pt/C, and commercial Pt were determined to be 35%, 25%, 30%, 30% and 20% by ICP-OES, respectively. We first evaluated the electrocatalytic performance of Pt/C and Pt_n_Co_100−n_/C for MOR. The MOR performance of Pt_n_Co_100−n_/C catalysts is shown in Figure 3a. The peak currents and peak potentials showed that the PtnCo100−n/C catalyst exhibited higher activity than the Pt/C catalyst. The Pt_53_Co_47_/C exhibited an MA of 2.15 A mg Pt^−1^, which was 1.38, 2.11, 3.58 and 4.78 times higher than those of Pt_27_Co_73_/C (1.56 A mg Pt^−1^), Pt_70_Co_30_/C (1.02 A mg Pt^−1^), Pt/C (0.60 A mg Pt^−1^), and commercial Pt/C (0.45 A mg Pt^−1^), respectively. The Pt_27_Co_73_/C exhibited the highest SA of 3.31 mA/cm^2^, which was 1.655, 2.98, 5.02 and 4.60 times higher than those of Pt_53_Co_47_/C (2.00 mA/cm^2^), Pt_70_Co_30_/C (1.11 mA/cm^2^), Pt/C (0.66 mA/cm^2^), and commercial Pt/C (0.72 mA/cm^2^), respectively (Figure 3b). Pt_27_Co_73_/C and Pt_53_Co_47_/C displayed higher activity for the MOR (Table 1) in comparison to the literature [32,33,34,35]. The EOR performance of commercial Pt/C and Pt _n_Co_100−n_/C catalysts is evaluated in Figure 3c,d. The Pt_n_Co_100−n_/C catalyst exhibited higher activity than commercial Pt/C, as shown in Figure 3c. Figure 3d and Table 2 show that Pt_27_Co_73_/C exhibited the largest MA value (2.11 A mg^−1^) and largest SA (1.44 mA/cm^2^) of EOR, which was 3.91 times and 1.67 times that of Pt/C (0.54 A mg^−1^and 0.86 mA/cm^2^), respectively. Pt_53_Co_47_/C and Pt_27_Co_73_/C had higher activity for the EOR (Table 3) compared with the literature [36,37,38,39]. From the catalytic performance of MOR and EOR, it can be seen that the introduction of Co played an important part in the improvement of their performance. The I_f_/I_b_ ratio can be used to certify the CO tolerance of catalysts [40]. The I_f_/I_b_ ratio of Pt_53_Co_47_/C catalyst was 1.21, larger than that of Pt/C catalysts (1.05) (Figure 3a), due to facilitating the oxidation of methanol via relieving the CO poisoning with Co. The I _f_/I _b_ ratio of the Pt_27_Co_73_/C catalyst was 1.14, larger than that of the Pt/C catalysts (0.83) (Figure 3c), which could be ascribed to its anti-poisoning enhancement in the presence of Co.

Figure 4a shows CVs of Pt_27_Co_73_/C, Pt_57_Co_43_/C, Pt_70_Co_30_/C and commercial Pt/C catalysts in 0.1 M HClO_4_ solution. As shown in Figure 4b, the electrochemical active surface area (ECSA) on behalf of the effective number of active catalytic sites, was determined from the H_2_ desorption peak areas of CV. The Pt_27_Co_73_/C, Pt_57_Co_43_/C, Pt_70_Co_30_/C and commercial Pt/C catalysts exhibited ECSA values of 30.6, 43.2, 54.9 and 49.9 m^2^g^−1^ Pt, respectively. The introduction of Co induced less exposure of the Pt active sites. However, the alloying with a higher atomic ratio of Co reduced the NW lengths and also led to an increase in the ECSA.

### 3.3. Durability

In practical applications, durability is also an important requirement for catalysts [41]. Hence, the initial stability of the Pt_n_Co_100−n_/C was assessed by chronoamperometric (CA) tests. Through a comparison of Pt_n_Co_100−n_/C catalyst with different components and commercial Pt/C, it was found that Pt_53_Co_47_/C and Pt_27_Co_73_/C have excellent catalytic activity. CA measurements of the catalysts were conducted at 0.65V (vs. SCE) in N_2_ -saturated 0.1 M HClO_4_ aqueous solution containing 0.5 M methanol/ethanol to evaluate their stability (Figure 4). CA curves indicated that the Pt_27_Co_73_/C catalyst and Pt_53_Co_47_/C electrocatalysts exhibited excellent stability. The Pt_27_Co_73_/C and Pt_53_Co_47_/C catalysts exhibited much higher initial current densities than Pt/C. The current density of Pt_27_Co_73_ C and Pt_53_Co_47_/C remaining were also much higher than that of Pt/C after 5000 s tests. There was about 22.1% and 21.8% activity retention of Pt_53_Co_47_ NWs/C and Pt_27_Co_73_ NWs/C, respectively, after 5000 s in MOR, much larger than that of commercial Pt/C (10% activity retention). There was about 26.1% and 24.8% activity retention of Pt_27_Co_73_ NWs/C and Pt_53_Co_47_ NWs/C, respectively, after 5000 s in EOR, much larger than that of commercial Pt/C (9% activity retention). The MOR current densities of Pt_53_Co_47_/C catalyst and Pt_27_Co_73_/C catalyst were 110.65 mA mg^−1^ and 59.92 mA mg^−1^ after 5000 s, which were 5.84 and 3.16 times higher than that of commercial Pt/C (18.95 mA mg^−1^), respectively (Figure 5a). The EOR current density of catalyst Pt_27_Co_73_/C (118.88 mA mg^−1^) and Pt_53_Co_47_/C (86.48 mA mg^−1^) catalyst were 5.47 and 3.98 times that of commercial Pt/C (21.73 mA mg^−1^) after 5000 s, respectively (Figure 5b). Moreover, compared with recently reported electrocatalysts [42,43,44,45,46,47], the Pt_53_Co_47_ NWs/C catalyst and Pt_27_Co_73_ NWs/C catalyst also exhibited excellent stability under the same conditions (Table 3 and Table 4). CA tests of the Pt_53_Co_47_/C for MOR were extended to 7500 s at 0.65 V vs. RHE. The current densities remained 91.4 mA mg^−1^ for MOR after 7500 s (Figure 5c). On the other hand, the electronic transfer rate was notably improved, due to the synergistic electronic interaction between Pt and Co atoms of Pt_n_Co_100−n_/C catalyst. As a consequence, these advantages enabled the catalyst to achieve superior catalytic performance and durability.

**Table 3 materials-16-00840-t003:** Stability comparison of Pt_53_Co_47_ NWs/C and Pt-based electrocatalysts for MOR.

Catalyst	Electrolyte	CA Stability (Activity Retention)	Potential	Reference
G@(PEI/Au) 3.5 @Pt	0.5 M H_2_SO_4_ + 1 M methanol	~5.4% after 2000 s	0.60 V (vs. SCE)	[42]
Pt_95_Co_5_ NWs		~7.0% after 3600 s	0.60 V (vs. SCE)	[43]
PdRuPt NWs	0.1 M HClO_4_ + 0.5 M methanol	~16.2% after 5000 s	0.60 V (vs. SCE)	[44]
Pt_53_Co_47_ NWs/C		~22.3% after 5000 s	0.65 V (vs. SCE)	**This work**

The excellent MOR and EOR activity and stability of Pt_n_Co_100−n_ NWs were mainly determined by the synergistic effect of Pt and Co bimetallic and the defect-rich nanowire structure. The ultrafine and ultrathin nanowire structure fully exposed the active sites. Ultrathin NWs with Boerdijk–Coxeter structure have different crystal plane orientations of the (111)-dominant facets that can control the electronic structure and generate lattice strain, which is beneficial to improve the performance of MOR and EOR.

## 4. Conclusions

In conclusion, we demonstrated a simple one-step hydrothermal method for the synthesis of ultrathin Pt_n_Co_100−*n*_ NWs with adjustable compositions. With lattice strain and dominating (111) facets, ultrathin PtnCo_100−_*_n_* NWs demonstrated significant MOR and EOR activity and stability while also enhancing alcohol oxidation catalysis. Pt_n_Co_100−_*_n_* NWs with controlled compositions have a low starting potential and enhanced MOR and EOR activity because of the robust electronic interaction between metals. The Pt_53_Co_47_/C exhibited the highest MA of 2.15 A mg Pt^−1^, which was 1.38-fold and 4.78-fold higher than that of Pt_27_Co_73_/C and commercial Pt/C for MOR. The Pt_27_Co_73_/C exhibited the highest SA of 3.31 mA/cm^2^, which was 1.655-fold and 4.60-fold higher than that of Pt_53_Co_47_/C and commercial Pt/C for MOR. The Pt_27_Co_73_/C exhibited the largest MA value (2.11 A mg^−1^) and largest SA (1.44 mA/cm^2^) of EOR, which were about 3.91-fold and 1.67-fold that of Pt/C, respectively. The current densities of Pt_27_Co_73_/C and Pt_53_Co_47_ NWs/C were much higher than that of commercial Pt/C after 5000 s tests. This study provides an ideal strategy for adjusting the composition of Pt-based alloys. We believe that this study will offer good insights for the preparation of fuel cell electrocatalysts with excellent performance and remarkable durability and will promote the future development of fuel cell electrocatalysts for energy conversions.

## Figures and Tables

**Figure 1 materials-16-00840-f001:**
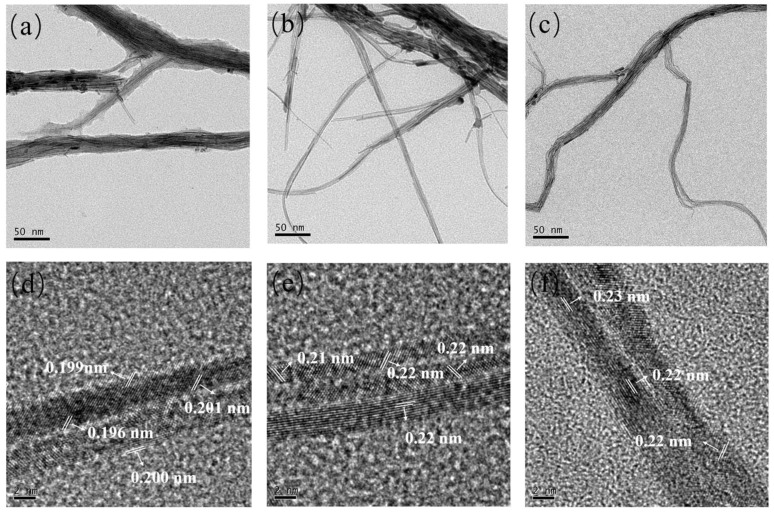
TEM images of Pt_27_Co_73_ (**a**), Pt_53_Co_47_ (**b**), Pt_70_Co_30_ (**c**). HR-TEM images of Pt_27_Co_73_ (**d**), Pt_53_Co_47_ (**e**) and Pt_70_Co_30_ (**f**) with lattice fringes.

**Figure 2 materials-16-00840-f002:**
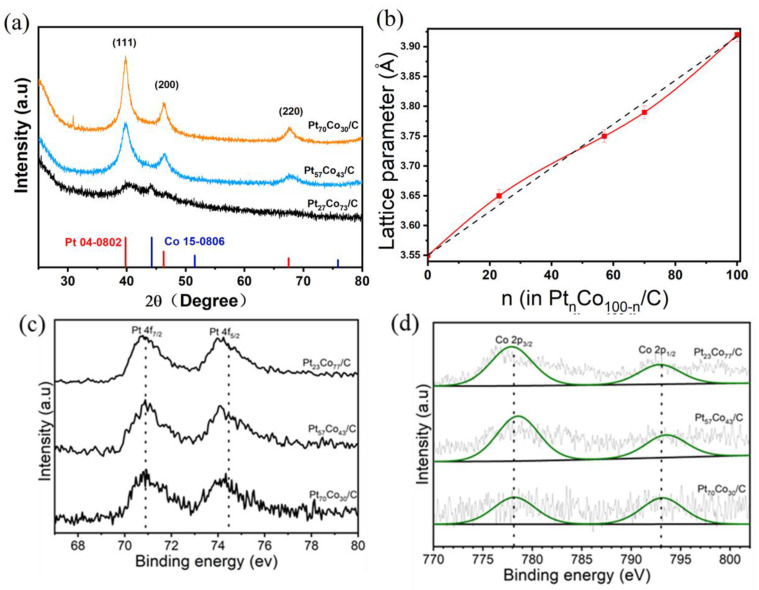
(**a**) XRD patterns of NWs. (**b**) Lattice parameters for NWs on the relative composition of Pt%. XPS spectra and deconvoluted peaks in the regions of (**c**) Pt 4f and (**d**) Co 2p. The international standard (C 1s) was used to calibrate the peak position.

**Figure 3 materials-16-00840-f003:**
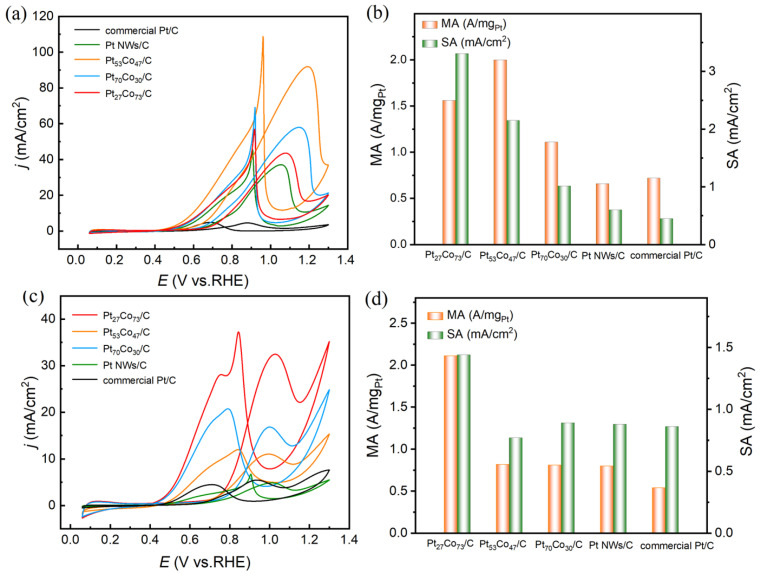
(**a**,**c**) CV curves of Pt_27_Co_73_/C, Pt_53_Co_47_/C, Pt_70_Co_30_/C, Pt/C and commercial Pt/C in 0.1 M HClO_4_ + 0.5 M CH_3_OH/C_2_H_5_OH solution purged with N_2_-saturated solution; (**b**,**d**) mass activity and specific activity data of methanol oxidation and ethanol oxidation.

**Figure 4 materials-16-00840-f004:**
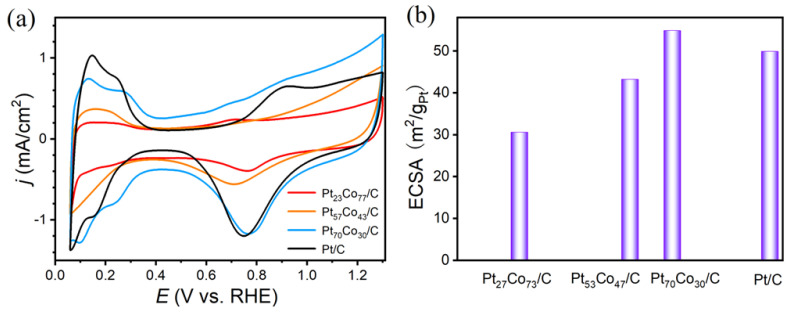
CV curves (**a**) with the corresponding ECSA (**b**) of Pt_27_Co_73_/C, Pt_53_Co_47_/C, Pt_70_Co_30_/C and commercial Pt/C in 0.1 M HClO_4_ solution.

**Figure 5 materials-16-00840-f005:**
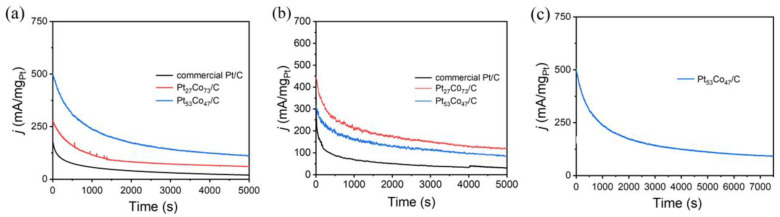
Electrocatalytic durability of the Pt_27_Co_73_/C, Pt_53_Co_47_/C and commercial Pt/C. Current-time curves of these catalysts recorded at 0.65 V. (**a**,**c**) chronoamperometric tests for MOR in 0.1 M HClO_4_ + 0.5 M methanol. (**b**) chronoamperometric tests for EOR in 0.1 M HClO_4_ + 0.5 M ethanol.

**Table 1 materials-16-00840-t001:** Comparison of MOR activities of various catalysts.

Catalyst	Electrolyte	MA (A/mg_Pt_)	SA (mA/cm^2^)	Reference
Pt_69_Ni_16_Rh_15_NWs/C	0.1 M HClO_4_ + 0.5 M methanol	1.72	2.49	[32]
UV-Pt@TONR/GN	0.5 M H_2_SO_4_ + 1 M CH_3_OH	1.94	3.16	[33]
Pd_9_Ru@Pt/FGN		0.881	-	[34]
1D PtFe alloy	0.1 M HClO_4_ + 0.5 M Methanol	1.65	-	[35]
Pt_27_Co_73_/C	0.1 M HClO_4_ + 0.5 M ethanol	1.56	3.31	**This work**
Pt_53_Co_47_/C		2.15	2.00	**This work**

**Table 2 materials-16-00840-t002:** Comparison of EOR activities of various catalysts.

Catalyst	Electrolyte	MA (A/mg_Pt_)	SA (mA/cm^2^)	Reference
PtSn/XC-72	0.5 M H_2_SO_4_ + 1 M ethanol	0.76	NA	[36]
Pt-Mo-Ni NWs	0.5 M H_2_SO_4_ + 2 M ethanol	0.87	2.57	[37]
PtRhNi/C		0.34	NA	[38]
PtCu_2.1_ NWs		1.02	2.16	[39]
Pt_27_Co_73_/C	0.1 M HClO_4_ + 0.5 M ethanol	2.11	1.44	**This work**
Pt_53_Co_47_/C		0.82	0.77	**This work**

**Table 4 materials-16-00840-t004:** Stability comparison of Pt_27_Co_73_ NWs/C and Pt-based electrocatalysts for EOR.

Catalyst	Electrolyte	CA Stability (Activity Retention)	Potential	Reference
Pt HCCLV	0.5 M H_2_SO_4_ + 1 M ethanol	~27.0% after 2000 s	0.60 V (vs. Ag/AgCl)	[45]
Pt_3_Sn/GO		~21.7% after 3000 s	peak potential	[46]
Octahedral Pt_2.3_Ni/C	0.1 M HClO_4_ + 0.5 M methanol	~14.7% after 1800 s	~0.63 V (vs. RHE)	[47]
Pt_27_Co_73_ NWs/C		~25.7% after 5000 s	0.65 V (vs. SCE)	**This work**

## Data Availability

Not applicable.

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
