# Peer review of "Platinum-Cobalt Nanowires for Efficient Alcohol Oxidation Electrocatalysis"

_materials, 2023, doi:10.3390/ma16020840_

Round 1

Reviewer 1 Report

1.   Shorter title is to be modified to show the purpose of the work.

2.   Abstracts need to be improved to present this study's significant outcome and innovation points.

3.    In your paper, please highlight the following: why this work has been done, what is new about it, highlight the novelty (has such work been done before, if so, why you are doing this work)

4.  The introduction provided general information and a literature survey on materials. There are diverse reports in the literature on materials obtained from these bimetallic alloys. The introduction lacks literature about similar materials that were employed.

5.    Figure 1- Hard to read the figure insets. Not readable.

6.    Figure X needs to be discussed uniformly in the manuscript

7.   There are many errors in the paper, so the Authors are encouraged to review the form and the English of the manuscript.

8.  Many space errors/punctuation errors must be solved. The abbreviations should be checked in the manuscript and made clear.

9.    Figure 2a- Peaks must be identified. No diffraction planes are described here. What is line indicates? Must be included more information on this, need to be discussed in detail.

10.    Figure 3- Different colors have been used for preparing materials. Hard to differentiate the different colors. Uniformity needs to be maintained.

11.    When presenting the results, compare electrochemical performance with the latest literature. After showing the results, it is recommended to provide a short conclusion to the obtained results.

12.    The conductivity of the prepared materials needs to be investigated.

13.    Conclusion section- must focus on future directions of these prepared materials?

Author Response

 Point-by-point responses and revisions made in the manuscript (Materials-2124777)

Reviewers' comments:

  1. Shorter title is to be modified to show the purpose of the work.

Reponses:

Thank you for your suggestion. The title has been revised in shorter in the revised manuscript. Thanks.

  1. Abstracts need to be improved to present this study's significant outcome and innovation points.

Reponses:

Thanks for your comments. We have revised the abstract to improve the study's significant outcome and innovation points. Thanks.

  1. In your paper, please highlight the following: why this work has been done, what is new about it, highlight the novelty (has such work been done before, if so, why you are doing this work)

Reponses:

Thank you for your comments. This work shows that the electrocatalytic performance of ultrathin PtnCo100-n nanowires (NWs) (≈2.1 nm ) is improved by regulating the compositions, high active facets and lattice strain of catalysts for alcohol oxidation. We have highlighted this novelty in the revised manuscript. Thanks.

  1. The introduction provided general information and a literature survey on materials. There are diverse reports in the literature on materials obtained from these bimetallic alloys. The introduction lacks literature about similar materials that were employed.

Reponses:

We sincerely appreciate the valuable comments. We have checked the literature carefully and added more references on PtCo bimetallic alloys into the introduction part in the revised manuscript. Thanks.

  1. Figure 1- Hard to read the figure insets. Not readable.

Reponses:

Thanks for your comments. We have modified it in the revised manuscript. Thanks.

  1. Figure X needs to be discussed uniformly in the manuscript.

Reponses:

Thanks for your comments. We have discussed uniformly Figure in the revised manuscript. Thanks.

  1. There are many errors in the paper, so the Authors are encouraged to review the form and the English of the manuscript.

Reponses:

Thanks for your comments.  We tried our best to improve the manuscript and made some changes to the manuscript. These changes will not influence the content and framework of the paper. We appreciate for reviewers' warm work earnestly and hope that correction will meet with approval. Thanks.

  1. Many space errors/punctuation errors must be solved. The abbreviations should be checked in the manuscript and made clear.

Reponses:

Thanks for your comments.  In our revised manuscript, the space errors/punctuation errors are revised. Thanks.

  1. Figure 2a- Peaks must be identified. No diffraction planes are described here. What is line indicates? Must be included more information on this, need to be discussed in detail.

Reponses:

Thanks for your comments. We have identified and added corresponding diffraction planes in Figure 2a. The red line indicates Pt (JCPDS No. 04-0802) and the blue line indicates Co (JCPDS No. 15-0806). We have added the suggested content to the manuscript. Thanks.

  1. Figure 3- Different colors have been used for preparing materials. Hard to differentiate the different colors. Uniformity needs to be maintained.

Reponses:

This suggestion is appreciated. This has been improved in the revised version. Thanks.

  1. When presenting the results, compare electrochemical performance with the latest literature. After showing the results, it is recommended to provide a short conclusion to the obtained results.

Reponses:

Thanks for your suggestion. We have compared the electrochemical performance of this work with the recent literature and provided a short conclusion to the obtained results. Please see Table 1-Table 4 in the revised manuscript. Thanks.

  1. The conductivity of the prepared materials needs to be investigated.

Reponses:

Thanks for your suggestion. We have added a section on electrochemical active surface area. Please see Fig. 4 in the revised manuscript. Thanks.

  1. Conclusion section- must focus on future directions of these prepared materials?

Reponses:

Thanks for your comments. This study provides an ideal strategy for adjusting composition of Pt-based alloys. We believe that this study will offer good insights for the preparation of fuel cell electrocatalysts with excellent performance and remarkable durability and will promote the future development of fuel cell electrocatalysts for energy conversions. We have revised the conclusion in the manuscript. Thanks.

Reviewer 2 Report

- For the benefit of the reader and to enable other researchers to repeat the procedure, some experimental details should be specified. What was the range of the Pt and Co precursors added and what was the amount of NW and Vulcan mixed together? What was the volume of ink deposited on the electrode?

 - The authors mention correlation between structural properties and electrocatalytic activity in the abstract, but there is no discussion on this correlation in the text itself. This is the main flaw of the manuscript and discussion should be added.

 - I would suggest to use the same colour code for MA and SA in figures 3 b and 3d.

 - Analysis of CO tolerance based on the ratio of forward and backward peak current densities could be useful for the reader. See for example J. Electroanal. Chem., 573 (2004), pp. 197-202 or Electrochem. Commun., 8 (2006), pp. 499-504.

 - The durability tests are rather short so I would suggest to refer to as “initial stability tests” or to run them for longer time than 2 h.

 - In stability tests it would be useful to compare the initial current density and current density at the end of the measurement for each catalyst.

 - What is the reason for the initial decay of current density?

Author Response

Point-by-point responses and revisions made in the manuscript (Materials-2124777)

Reviewers' comments:

Comment 2

For the benefit of the reader and to enable other researchers to repeat the procedure, some experimental details should be specified. What was the range of the Pt and Co precursors added and what was the amount of NW and Vulcan mixed together? What was the volume of ink deposited on the electrode?

Reponses:

Thanks for your comments.  In our revised manuscript, some experimental details have been specified. Thanks.

  1. The authors mention correlation between structural properties and electrocatalytic activity in the abstract, but there is no discussion on this correlation in the text itself. This is the main flaw of the manuscript and discussion should be added.

We sincerely appreciate the valuable comments. As your suggestion, we have added the discussion about correlation between structural properties and electrocatalytic activity into the durability part in the revised manuscript. Thanks.

  1. I would suggest to use the same colour code for MA and SA in figures 3 b and 3d.

Reponses:

Thanks for your comments. We have use the same colour code for MA and SA in figures 3 b and 3d. Thanks.

  1. Analysis of CO tolerance based on the ratio of forward and backward peak current densities could be useful for the reader. See for example J. Electroanal. Chem., 573 (2004), pp. 197-202 or Electrochem. Commun., 8 (2006), pp. 499-504.

Reponses:

Thanks for your suggestion. We have discussed the CO tolerance based on the ratio of forward and backward peak current densities according to your suggestion, and added the suggested content to the manuscript. Thanks.

  1. The durability tests are rather short so I would suggest to refer to as “initial stability tests” or to run them for longer time than 2 h.

Reponses:

Thanks for your comments. As suggested by the reviewer, we have improved this in the revised version. Thanks.

  1. In stability tests it would be useful to compare the initial current density and current density at the end of the measurement for each catalyst.

Reponses:

Thanks for your comments. We have added the content about comparing the initial current density and current density at the end of the measurement for catalysts into the durability part in the revised manuscript. Thanks.

  1. What is the reason for the initial decay of current density?

Reponses:

Thanks for your comments. We believe that one reason of the initial decay of current density is the loss of Co from the catalysts, which may mean a change in catalysts composition. Thanks.

Round 2

Reviewer 1 Report

Accept in present form